# Estimation of Nuclear DNA Content in Some *Aegilops* Species: Best Analyzed Using Flow Cytometry

**DOI:** 10.3390/genes13111980

**Published:** 2022-10-29

**Authors:** Solmaz Najafi, Mehmet Ulker, Erol Oral, Ruveyde Tuncturk, Murat Tuncturk, R. Z. Sayyed, Kahkashan Perveen, Peter Poczai, Andras Cseh

**Affiliations:** 1Department of Field Crops, Faculty of Agriculture, Van Yuzuncu Yil University, Van 65090, Turkey; 2Department of Microbiology, PSGVP Mandal’s S I Patil Arts, G B Patel Science and STKV Sangh Commerce College, Shahada 425409, India; 3Department of Botany & Microbiology, College of Science, King Saud University, P.O. Box 22452, Riyadh 11495, Saudi Arabia; 4Finnish Museum of Natural History, University of Helsinki, FI-00014 Helsinki, Finland; 5Agricultural Institute, Centre for Agricultural Research, ELKH, 2462 Martonvásár, Hungary

**Keywords:** *Aegilops*, genome size, nuclear DNA content, ploidy level

## Abstract

The genera *Triticum* and *Aegilops* have been considered as the main gene pool of wheat due to their features, such as tolerance of all types of abiotic and biotic stresses. This study was conducted to evaluate the cytogenetic analyses in 115 native and wild populations from eleven *Aegilops* species using their nuclear DNA quantification. Mean 2C nuclear DNA contents of different ploidy levels in the wild wheat of Turkey and Iran were measured using the flow cytometry technique. The obtained results showed that the mean nuclear DNA content in diploid species varied from 10.09 pg/2C (*Ae. umbellulata*) to 10.95 pg/2C (*Ae. speltoides* var. *ligustica*) in Turkey. In Iranian diploids, the mean nuclear DNA content varied from 10.20 pg/2C (*Ae. taushii*) to 11.56 pg/2C (*Ae. speltoides* var. *ligustica*). This index in the tetraploid species of Turkey varied from 18.09 pg/2C (*Ae. cylindrica*) to 21.65 pg/2C (*Ae. triaristata*), and in Iranian species, it was from 18.61 pg/2C (*Ae. cylindrica*) to 21.75 pg/2C (*Ae. columnaris*). On the other hand, in the hexaploid species of Turkey, this index varied from 31.59 pg/2C (*Ae. crassa*) to 31.81 pg/2C (*Ae. cylindrica*); in the Iranian species, it varied from 32.58 pg/2C (*Ae. cylindrica*) to 33.97 pg/2C (*Ae. crassa*). There was a significant difference in the DNA content of Turkey and Iran diploid as well as tetraploid species; however, in hexaploid species, the difference was not significant. It was concluded that the variation in intraspecific genome size was very low in diploid and tetraploid populations; this means that the low variation is not dependent on geographic and climatic parameters. On the other hand, the interspecific variation is significant at the diploid and tetraploid populations. It is generally very difficult to distinguish *Aegilops* species from each other in natural conditions; meanwhile, in this study, all species could be, easily, quickly and unambiguously, distinguished and separated using the FCM technique.

## 1. Introduction

Wheat) *Triticum aestivum* L.) is the most important grain crop worldwide. It is produced in a wide range of climatic conditions and geographical areas; due to its high adaptation to water as well as the global production of more than 700 million tons, this crop provides 20% of the daily protein and caloric needs of 4.5 billion people worldwide [1]. The development of global climate change, genetic erosion and the challenge of sustainable agricultural production have highlighted the need to exploit heritage resources, especially wild relatives [2]. Wheat wild relatives are candidate gene reservoirs with potential use in the genetic improvement of wheat [3]. Wild relatives of wheat contain beneficial genes, such as those associated with resistance to a variety of biological and non-biological stresses [4]. Irrigation always plays an important role in increasing the yield of most crops [5]. It has been reported that the type and method of irrigation can increase the wheat and corn yield by 35% and 23%, respectively. In addition, the hydrological cycle between land and atmospheric can be effective on irrigation [6]. One of the most prominent and well-known main irrigation methods in the world for wheat to increase crop productivity is sprinkler irrigation [7,8,9]. According to the data obtained from the International Commission on Irrigation and Drainage, sprinkler-irrigated areas account for 40% and 10% of the total water surface in developed and developing countries, respectively [10]. Hence, the presence of such genetic resources can be used as useful germplasms in wheat breeding programs [11].

*Aegilops* L. is a genus of wheat mostly distributed in areas such as Mediterranean Sea and Asian countries such as Turkey and Iran [12,13]. Different species of *Aegilops* can be mixed with each other and with different species of *Triticum* [12,14]. It is not easy to identify different *Aegilops* species due to their extensive morphological similarities and gene flow among them, as well as cultivated wheat. It has been reported that around 21 *Aegilops* species (11 diploid, 10 tetraploid and 2 hexaploid) belong to six sections [13,15]. The *Aegilops* species have an important role in the evolution as well as improvement of the genetic variation process in cultivated common wheat (*T. aestivum* L.) [16]. The wild ancestors of wheat have a higher grain protein content when compared to modern wheat cultivars, which could be a source of protein transfer to wheat [17]. Nearly 200 wheat-*Aegilops* interspecific hybrids and translocation lines have been developed and almost 53 disease and insect resistance genes from 15 *Aegilops* species have been incorporated into the wheat gene pool [18]. Different species of the genus *Triticum* and *Aegilops* are the most important wild relatives of crop, which have been directly or indirectly introduced as B, A and D genomes [19,20]. For example, optimal and complete growth of *Ae. tauschii* in low rainfall areas with an average annual rainfall of 150 to 350 mm has caused this species to be considered as one of the species with drought tolerance genes [21].

Determination of DNA contents per nucleus, chromosome number, and morphological characteristics are important in plant breeding programs. Nuclear DNA content (C value) is the nuclear DNA quantity in the genome of any organism [22], regardless of taxon’s ploidy level. It is also an important variable in determining the morphology, biology, ecology and plant distribution. The intra- and inter-specific variation of nuclear DNA content in 43 *Aegilops* accessions has been reported [23]. The content of nuclear DNA in *Aegilops* species including *A. markgrafii* (Greuter) Hammer, *A. sharonensis* Eig., *A. geniculate* Roth and *A. neglecta* Req. ex Bertol was measured to be 4.84 pg, 7.52 pg, 9.23 pg and 16.35 pg, respectively [24,25,26]. In addition, the DNA content of *T. timopheevii* is reported less than that of *T. turgidum* L. [27,28]. The DNA content of *T. timopheevii* [27], *T. araraticum* [29] and *T. durum* Desf. [30] has been been measured to be 11.30 pg, 10.05 pg and 12.28 pg, respectively. In *T. araraticum*, *T. timopheevii*, *T. dicoccoides* and *T. durum* species, the nuclear DNA contents were measured to be 11.8 pg, 11.87 pg, 12.84 pg and 12.91 pg, respectively [31]. Flow cytometry is an important method in basic cell biology, which implies passing a visible spectrum of light in a short time through a small region of single particles suspended in a liquid to detect various chemical or biological components. Evaluation of large number of particles in a very short time is the most important advantage of this technique. Ploidy determination has been traditionally carried out by counting stained root tips chromosomes; however, it is usually time consuming and needs experiences, as well as tissues with dividing cells, especially in plant species with small chromosomes and high levels of ploidy [32]. The traditional method for nuclear DNA content measurement is the Feulgen micro spectrophotometry of root tips or shoots tips’ mitotic cells [30]. Flow cytometry is an easier, quicker, and more accurate method for nuclear DNA content estimation [33,34]. By this technique, it has become feasible and practical to screen large populations for the desired cytotypes, as well as the ploidy level characterization of plant materials, especially those kept in gene banks. There are several studies on the determination of nuclear DNA contents using flow cytometry in major crop plant species, as well as 13 turf grass species [35], perennial *Triticeae* [36], switchgrass (*Panicum virgatum* L.) [37,38] and alfalfa (*Medicago sativa* L.) [32]. The aim of this study was to, therefore, characterize the genome of diploid, tetraploid and hexaploid *Aegilops* using both cytogenetic and flow cytometry methods. For this purpose, we studied diploid, tetraploid and hexaploid wild wheat populations collected from Turkey and Iran.

## 2. Materials and Methods

### 2.1. Experimental Materials

During the 2019–2020 period, five different populations of each *Aegilops* species were collected from different regions in Turkey and Iran based on eco-geographic surveys. About 115 accessions representing 4 diploids (*Ae*. *speltoides* var. aucheri, *Ae*. *speltoides* var. ligustica, *Ae. tauschii* and *Ae. umbellulata*), 6 tetraploids (*Ae*. *biuncialis*, *Ae*. *columnaris*, *Ae*. *crassa* (4x), *Ae. cylindrica* (4x), *Ae. triaristata* and *Ae. triuncialis*) and 2 hexaploid (*Ae. crassa* (6x) and *Ae. cylindrica*(6x)) from Turkey (Table 1) and 4 diploids (*Ae*. *speltoides* var. *aucheri, Ae*. *speltoides* var. *ligustica*, *Ae. tauschii* and *Ae. umbellulata*), 5 tetraploids (*Ae*. *biuncialis*, *Ae*. *columnaris*, *Ae*. *crassa* (4x), *Ae. cylindrica* (4x) and *Ae. triuncialis*) and 2 hexaploid (*Ae. crassa* (6x) and *Ae. cylindrica* (6x)) from Iran (Table 1) were collected. Then, the samples were planted in greenhouse conditions for morphological characterization” and DNA extraction was used for the determination of DNA content per nucleus by the flow cytometry technique.

### 2.2. Chromosome Analysis

For karyotype analysis, root tip meristems were pretreated with 8-Hydroxiquinolin solution; then, they were fixed in the Lewitsky solution. The treated root tips were rinsed by distilled water and hydrolyzed for 10 min at 60 °C in HCl (1N); this was followed by staining in Aceto-Orcein. Ten metaphasic plates were used for analysis. Microscopic slides were made using the squash method for measuring several chromosomal features including somatic chromosome number, mean total chromosomes length (MTCL), mean arm ratio (MAR) and karyotype formula in each species, by using the photographic prints magnified to 3200×.

### 2.3. Determination of Nuclear DNA Content

This study was carried out using the Flow Cytometer (PARTEC, CyFlow Space, Nürnberg, Germany) at the Plant Genetics and Cytogenetics Laboratory, Department of Field Crops, Faculty of Agriculture, Namık Kemal University, in Turkey.

Accordingly, freshly grinded *Aegilops* leaf tissues (as the test materials) and freshly grinded barley (*Hordeum vulgare*) leaflets (2n = 2x = 14, with the nuclear DNA content of 10.68 pg as the standard) were cut followed by placing between wet filter papers in Petri plates and then transferred to laboratory condition for nuclear DNA content analysis. The Propidium Iodide (PI) kit was used based on the proposed protocol as follows: the fresh and green leaf tissues (20 mg) of both sample and standard plants were mixed together on ice; this was followed by adding 500 μL of the extraction buffer, crushed using a sharp scalpel for 30–60 s and gently shaken for 10–15 s. The extraction was passed in tubes through 50 µm filters (CellTrics, PARTEC, Nürnberg, Germany) and then incubated at darkness by adding 2 mL of staining solution (Dapi) into the tubes for 30–60 min. The samples were then analyzed by Flow Cytometer.

The DNA content in a nucleus of specific plant could be determined by comparing with a standard plant whose DNA content is known. For this purpose, as mentioned before, the standard plant tissues (barley) were also prepared at the same time as for the sample tissues. The absolute DNA content of a sample (pg) was measured using the mean fluorescence values of the standard and sample plants G1 peaks [39,40,41] using the following formula
$$\mathrm{Nuclear}\;\mathrm{DNA}\;=\frac{\mathrm{Fluorescent}\;\mathrm{intensity}\;\mathrm{of}\;\mathrm{the}\;\mathrm{plant}\;\mathrm{sample}\;}{\mathrm{Fluorescent}\;\mathrm{intensity}\;\mathrm{of}\;\mathrm{standard}\;\mathrm{plant}\;\mathrm{samples}\;}\times \;\mathrm{standard}\;\mathrm{plant}\;\mathrm{DNA}\;\mathrm{content}\;\left(\mathrm{pg}\right)$$

## 3. Results

The number of somatic chromosomes, the mean total chromosome length, mean arm ratio, and karyotypic formula of all *Aegilops* ecotypes collected from different regions of Iran and Turkey were analyzed (Table 2 and Table 3).

As can be seen in Figure 1, Figure 2, Figure 3, Figure 4, Figure 5, Figure 6, Figure 7, Figure 8, Figure 9, Figure 10, Figure 11, Figure 12, Figure 13, Figure 14, Figure 15, Figure 16, Figure 17, Figure 18, Figure 19, Figure 20 and Figure 21, in both countries, the distribution of *Aegilops* species is shown on the map and the samples studied were collected from the same areas. In addition, the morphology of the spikelet of the same species has been determined. From each species, a metaphase plate is presented as a representative of that species along with the karyotype of that species (Figure 1, Figure 2, Figure 3, Figure 4, Figure 5, Figure 6, Figure 7, Figure 8, Figure 9, Figure 10, Figure 11, Figure 12, Figure 13, Figure 14, Figure 15, Figure 16, Figure 17, Figure 18, Figure 19, Figure 20 and Figure 21). In addition, genome size was measured by a flow cytometer. Thus, the device drew a histogram for each injected sample that had statistical information, such as the number of counted nuclei, average peak size and coefficient of variation (CV) (Figure 22). 

**Figure 1 genes-13-01980-f001:**
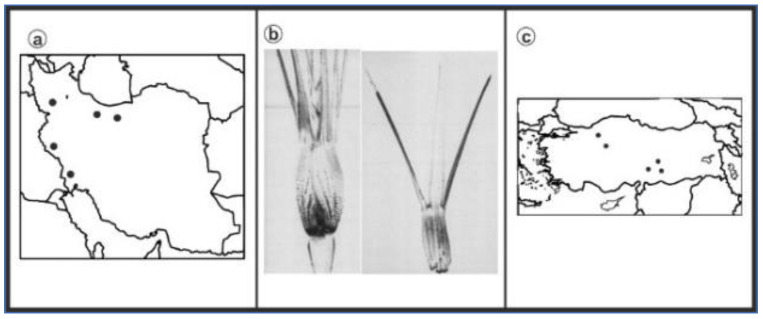
*Ae. biuncialis*, (**a**) Iran; (**b**) Spikelet; (**c**) Turkey.

**Figure 2 genes-13-01980-f002:**
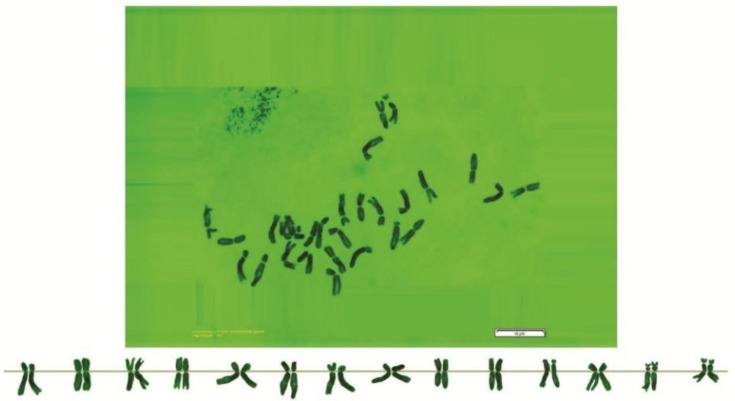
Microphotograph of somatic metaphases in *Ae. biuncialis*.

**Figure 3 genes-13-01980-f003:**
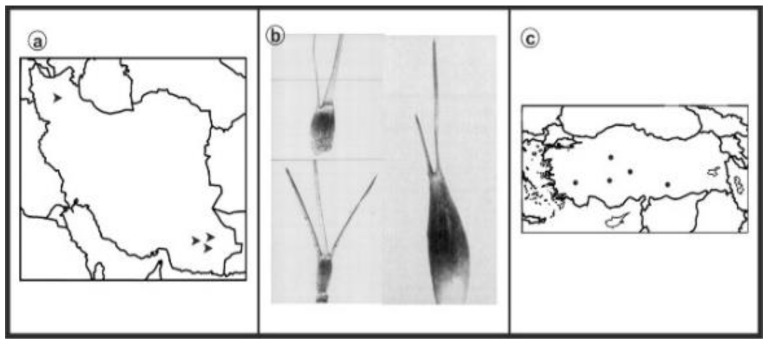
*Ae. columnaris*, (**a**) Iran; (**b**) Spikelet; (**c**) Turkey.

**Figure 4 genes-13-01980-f004:**
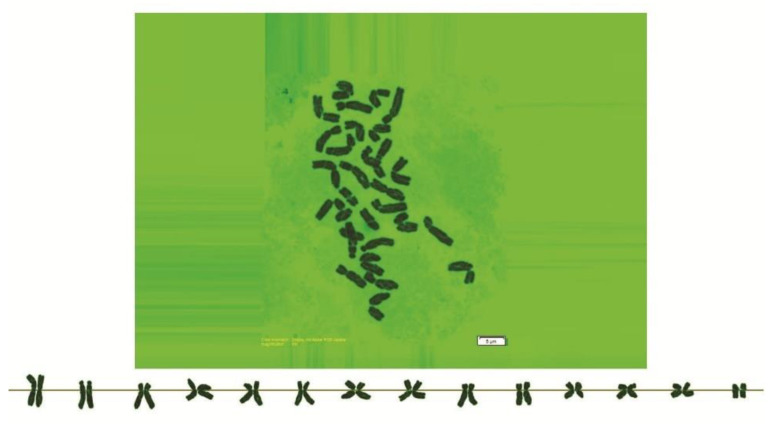
Microphotograph of somatic metaphases in *Ae. columnaris*.

**Figure 5 genes-13-01980-f005:**
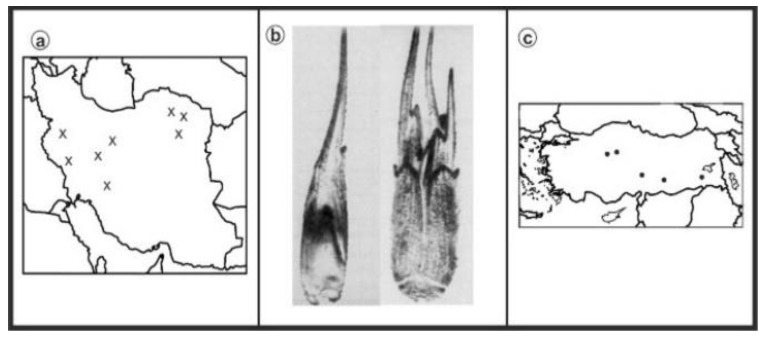
*Ae. Crassa* (4x, 6x), (**a**) Iran; (**b**) Spikelet; (**c**) Turkey.

**Figure 6 genes-13-01980-f006:**
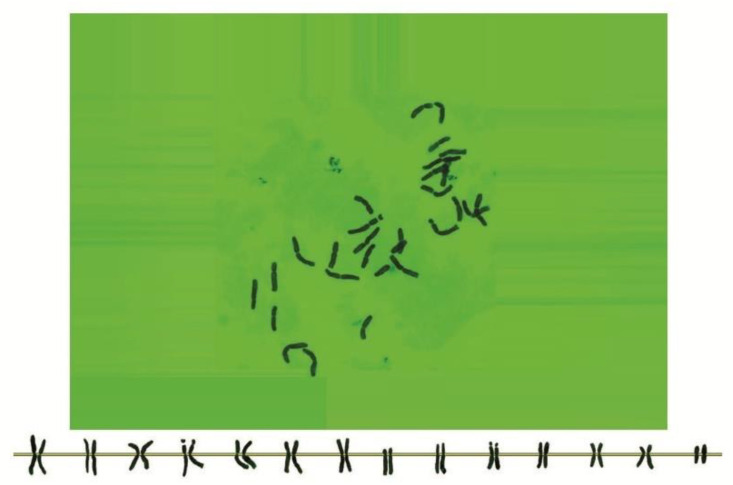
Microphotograph of somatic metaphases in *Ae. crassa* (4).

**Figure 7 genes-13-01980-f007:**
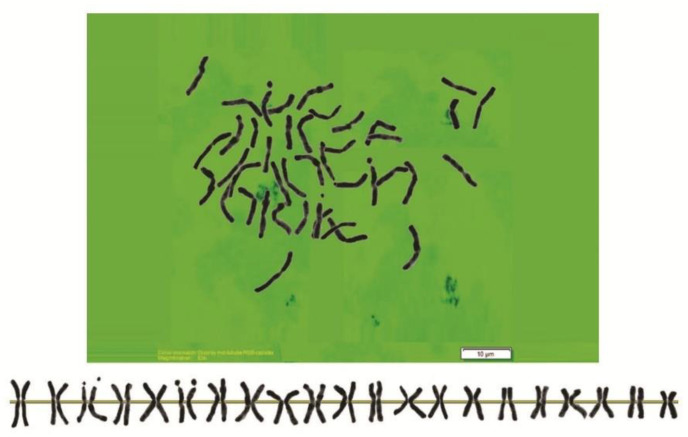
Microphotograph of somatic metaphases in *Ae. crassa* (6x).

**Figure 8 genes-13-01980-f008:**
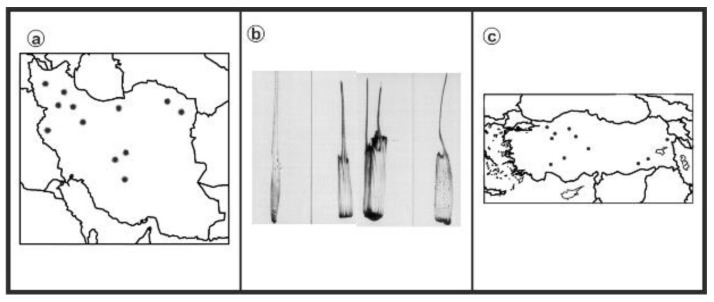
*Ae. cylindrica* (4x, 6x), (**a**) Iran; (**b**) Spikelet; (**c**) Turkey.

**Figure 9 genes-13-01980-f009:**
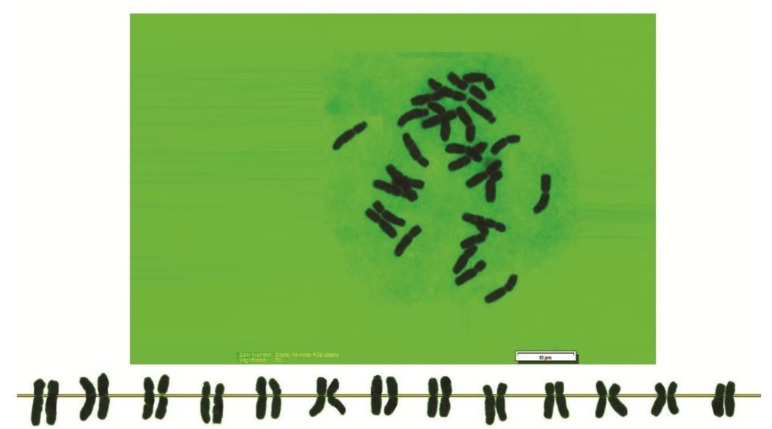
Microphotograph of somatic metaphases in *Ae. cylindrica* (4x).

**Figure 10 genes-13-01980-f010:**
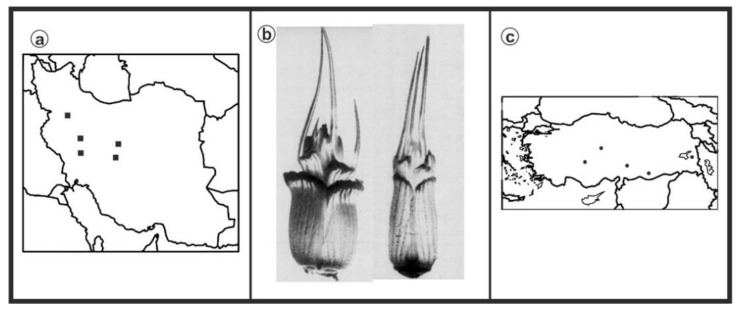
*Ae. speltoides* Taush, (**a**) Iran, (**b**) Spikelet, (**c**) Turkey.

**Figure 11 genes-13-01980-f011:**
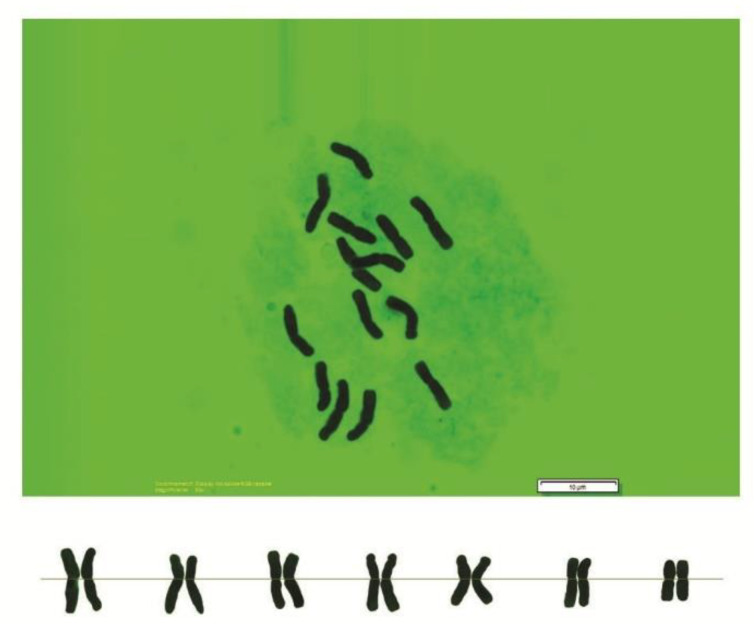
Microphotograph of somatic metaphases in Ae. speltoides Taush.

**Figure 12 genes-13-01980-f012:**
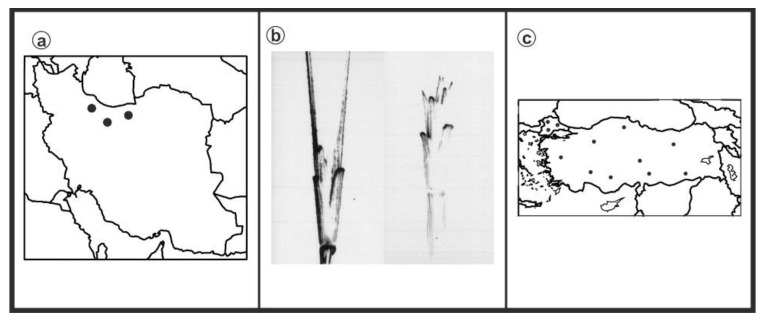
*Ae. speltoides* Var. aucheri, (**a**) Iran, (**b**) Spikelet, (**c**) Turkey.

**Figure 13 genes-13-01980-f013:**
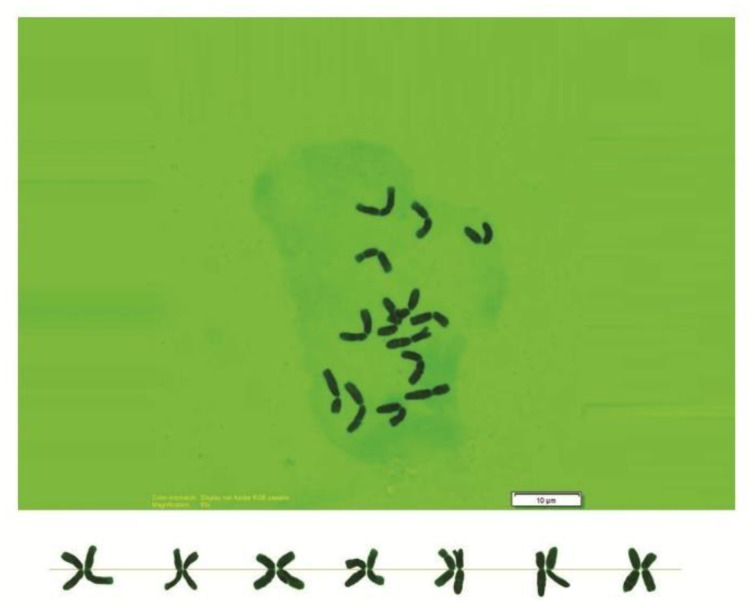
Microphotograph of somatic metaphases in *Ae. speltoides* Var. *aucheri*.

**Figure 14 genes-13-01980-f014:**
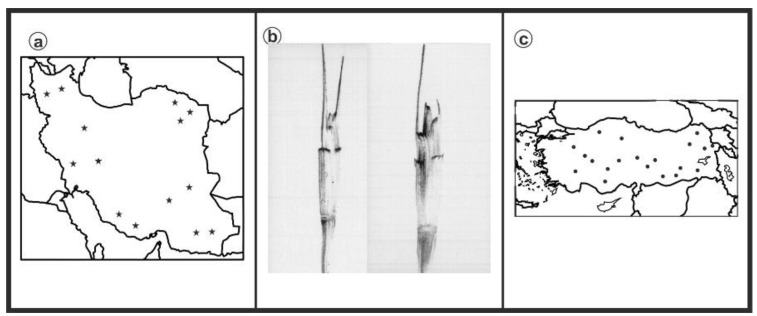
*Ae. speltoides* Var. ligustica, (**a**) Iran, (**b**) Spikelet, (**c**) Turkey.

**Figure 15 genes-13-01980-f015:**
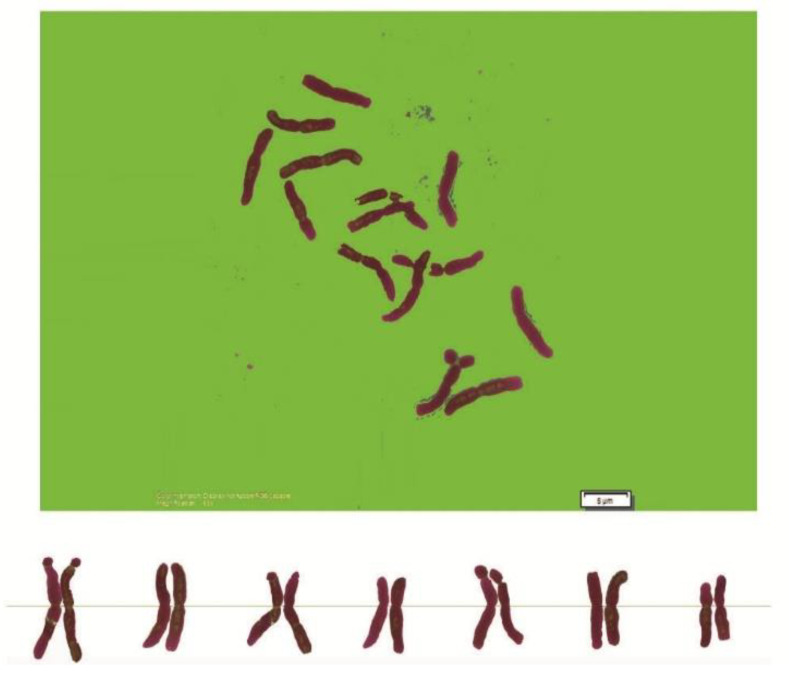
Microphotograph of somatic metaphases in *Ae. speltoides* Var. *ligustica*.

**Figure 16 genes-13-01980-f016:**
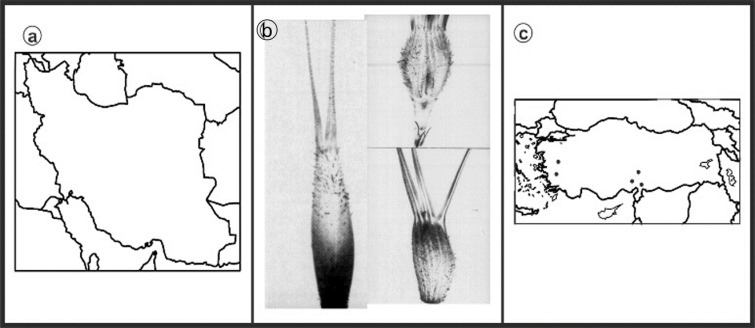
*Ae. triaristata*, (**a**) Iran, (**b**) Spikelet, (**c**) Turkey.

**Figure 17 genes-13-01980-f017:**
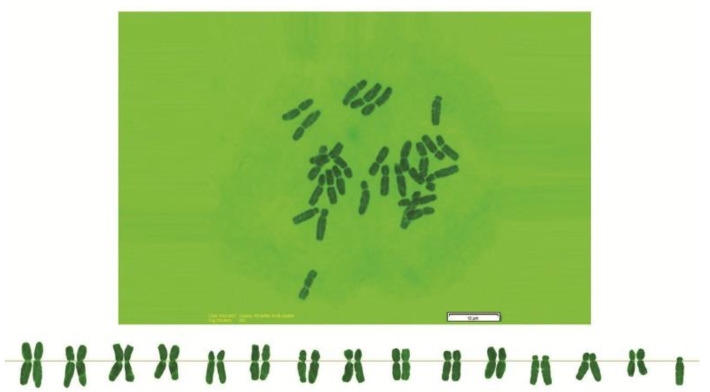
Microphotograph of somatic metaphases in *Ae. triaristata*.

**Figure 18 genes-13-01980-f018:**
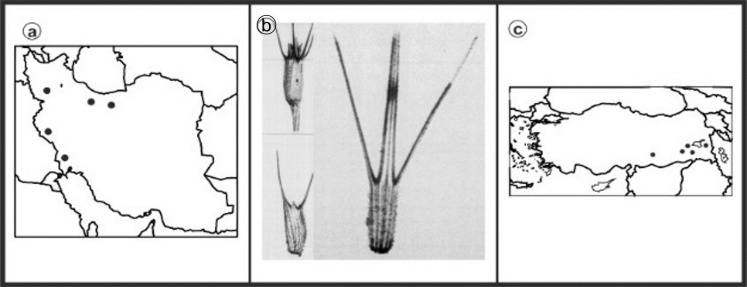
*Ae. triuncialis*, (**a**) Iran; (**b**) Spikelet; (**c**) Turkey.

**Figure 19 genes-13-01980-f019:**
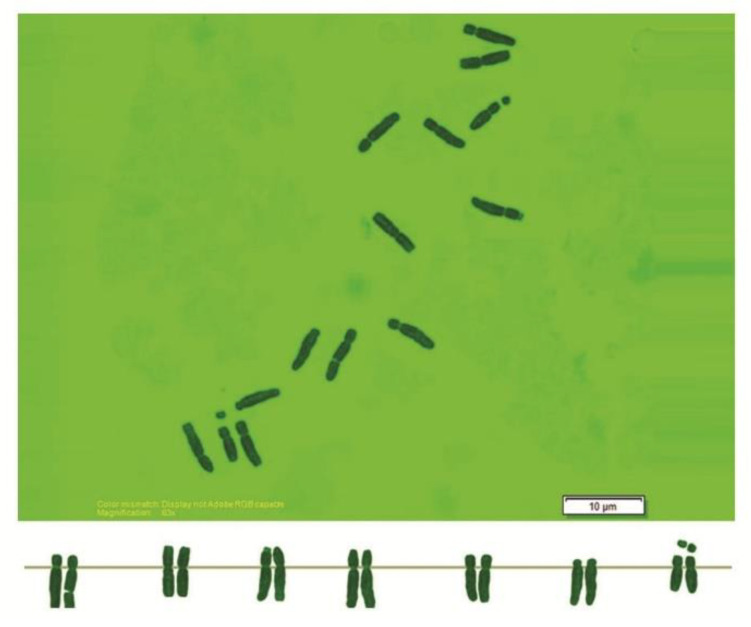
Somatic metaphases in *Ae. triuncialis*.

**Figure 20 genes-13-01980-f020:**
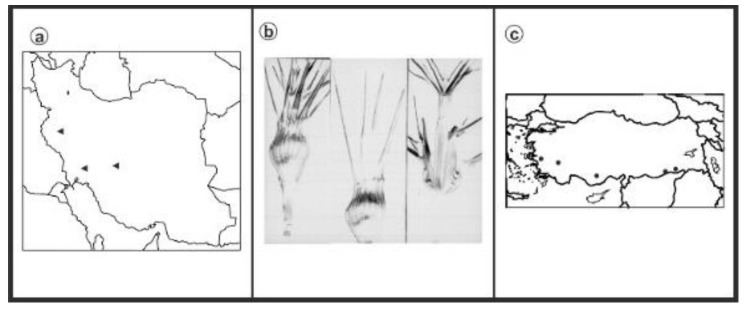
*Ae. umbellulata*, (**a**) Iran; (**b**) Spikelet; (**c**) Turkey.

**Figure 21 genes-13-01980-f021:**
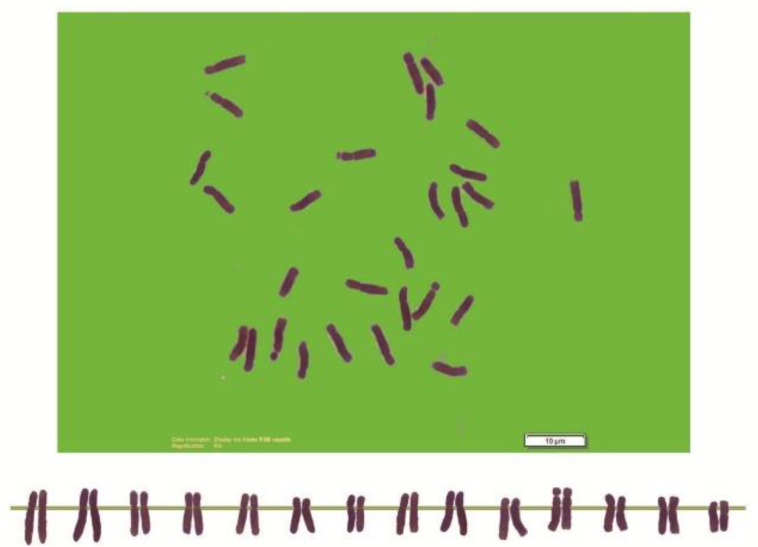
Somatic metaphases in *Ae. umbellulate*.

**Figure 22 genes-13-01980-f022:**
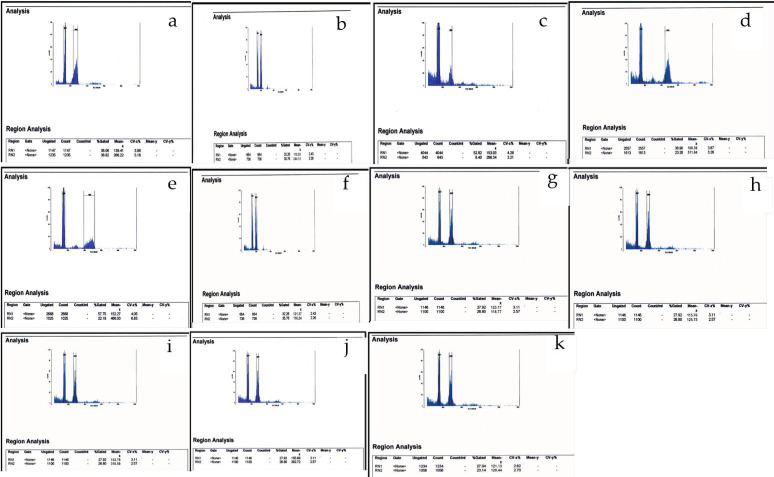
Peaks obtained from flow cytometry in *Aegilops* species; (**a**) *Ae. Biuncialis*; (**b**) *Ae. Columnaris*; (**c**) *Ae. crassa* (4x); (**d**) *Ae. crassa* (6x); (**e**) *Ae. Cylindrica*; (**f**) *Ae. speltoides* Taush; (**g**) *Ae. speltoides* var. *aucheri*; (**h**) *Ae. speltoides* var. ligustica; (**i**) *Ae. Triaristata*; (**j**) *Ae. Triuncialis*; (**k**) *Ae. umbellulate*. Mean 2C nuclear DNA contents of the *Aegilops* species in Turkey and Iran are indicated in Table 4. The nuclear DNA content mean in Turkey diploid species varied from 10.09 pg/2C (*Ae. umbellulata*) to 10.95 pg/2C (*Ae. speltoides* var. *ligustica*). In Iranian diploid species, the mean nuclear DNA content varied from 10.20 pg/2C (*Ae. speltoides* Taush) to 11.56 pg/2C (*Ae. speltoides* var. *ligustica*). Meanwhile, the nuclear DNA content mean in Turkey tetraploid species varied from 18.09 pg/2C (*Ae. cylindrica*) to 21.65 pg/2C (*Ae. triaristata*) and, in the Iranian species, it was from 18.61 pg/2C (*Ae. cylindrica*) to 21.75 pg/2C (*Ae. columnaris*). In addition, the nuclear DNA content mean in Turkey hexaploid species varied from 31.59 pg/2C (*Ae. crassa*) to 31.81 pg/2C (*Ae. cylindrica*); in Iranian species, it ranged from 32.58 pg/2C (*Ae. cylindrica*) to 33.97 pg/2C (*Ae. crassa*).

**Table 4 genes-13-01980-t004:** The mean of nuclear DNA content in *Aegilops* species.

Species	x	Genome ^1^	DNA Content (pg/2C) *	DNA Content (pg/2C) **	DNA Content (pg/2C) ***	DNA Content (pg/2C) ****	DNA Content (pg/2C) *****
*Ae. speltoides* Var. *aucheri*	2n = 2x = 14	SS	-	-	-	10.15 ± 0.04	10.22 ± 0.02
*Ae. speltoides* Var. *ligustica*	2n = 2x = 14	SS	-	-	-	10.95 ± 0.12	11.56 ± 0.01
*Ae. speltoides* Var. *Taush*	2n = 2x = 14	SS	10.20 ± 0.09	10.2	10.34 ± 0.08	10.25 ± 0.27	10.36 ± 0.36
*Ae. umbellulata*	2n = 2x = 14	UU	10.59 ± 0.11	10.1	10.76 ± 0.07	10.09 ± 0.04	10.58 ± 0.56
*Ae. biuncialis*	2n = 4x = 28	UUMM	20.61 ± 0.22	22.6	20.74 ± 0.04	19.89 ± 0.29	20.23 ± 0.35
*Ae. columnaris*	2n = 4x = 28	UUMM	21.75 ± 0.20	21.0	21.72	20.17 ± 0.16	21.75 ± 0.36
*Ae. crassa* (4x)	2n = 4x = 28	DDMM	21.29 ± 0.24	20.9	21.72	20.08 ± 0.01	20.44 ± 0.29
*Ae.cylindrica* (4x)	2n = 4x = 28	CCDD	18.79 ± 0.09	9.3	19.18	18.09 ± 0.05	18.61 ± 0.07
*Ae. triaristata*	2n = 4x = 28	UUMM	21.87 ± 0.25	31.0	21.28 ± 0.40	21.65 ± 0.05	-
*Ae. triuncialis*	2n = 4x = 28	UUCC	19.40 ± 0.17	18.9	19.86 ± 0.04	19.05 ± 0.03	19.27 ± 0.04
*Ae. crassa* (6x)	2n = 6x = 42	DDDDMM	33.63 ± 0.33	31.4		31.59 ± 0.03	33.97 ± 0.06
*Ae.cylindrica* (6x)	2n = 6x = 42	CCDDDD	-	-	-	31. 81 ± 0.08	32.58 ± 0.25

* Nuclear DNA content of Aegilops species taken from [42]; ** Nuclear DNA content of Aegilops species taken from the C-values database of [30]; *** Nuclear DNA content of Aegilops species taken from [26]; **** Nuclear DNA content of Aegilops species (TURKEY) observed in the current study; ***** Nuclear DNA content of Aegilops species (IRAN) observed in the current study; ^1^ Genome designations according to [43].

Results showed a significant difference in the DNA content of Turkey and Iran diploid, as well as tetraploid *Aegilops* species (*p* ˂ 0.01); however, in hexaploid species, a significant difference could not be observed (Table 5).

Somatic chromosomes, the mean total chromosome length, mean arm ratio, karyotypic formula, values of nuclear DNA content and ploidy levels of all *Aegilops* ecotypes collected from Iran and Turkey were analyzed (Table 6 and Table 7).

## 4. Discussion

Different ecotypes of *Aegilops* species used in this study (Table 1) showed that 12 and 11 species were collected from Turkey and Iran, respectively. *Ae. triaristata* species were only found in Turkey in Adana, Osmaniye, Tufanbeyli, Aydin and Manisa regions, but they could not be observed in Iran. The obtained results also showed that most *Aegilops* species in Turkey were observed in the Konya region; these included *Ae. speltpides* var *aucheri*, *Ae. speltpides* var. *ligustica*, *Ae. taushii, Ae. crassa* and *Ae. cylindrica* with two cytotypes (tetraploids, hexaploids) from *Ae. cylindrica* and *Ae. crassa*. The majority *Aegilops* species in Iran could be observed in the Shiraz region; these included *Ae. speltpides* var. *ligustica*, *Ae. taushii, Ae. biuncialis*, *Ae. crassa*, *Ae. cylindrica*, and *Ae. triuncialis* with two cytotypes (tetraploids, hexaploids) from *Ae. crassa* and *Ae. cylindrica*. In both Iran and Turkey, most collected *Aegilops* species were tetraploids followed by diploids and hexaploids. 

One of the most important applications of flow cytometry in plants has been the estimation of plant nuclear DNA content. Since this method measures the relative fluorescence intensity of nuclei stained by fluorochrome DNA, ploidy determination and estimation of nuclear DNA content in separate absolute units both require comparison with a reference standard of known DNA content; therefore, the quality of the obtained results depends on the selection and use of the standard. For accurate measurements in this method, the selection of a correct internal standard is required, so that nuclei of an unknown sample and a reference standard can be simultaneously isolated, stained and measured [44]. Flow cytometry works with determining light scattering and fluorescence of microscopic particles in the channel of movement of these particles at high speed in a narrow stream of liquid [45]. In plants, the most popular application of FCM has been the estimation of the nuclear DNA content (genome size, C-values) [46,47].

Determination of relative nuclear DNA values by flow cytometry could serve as a supplementary simple and routine method for the identification and maintenance of accessions [48]. An amount of nuclear DNA is applied to interpret the evolutionary relationships of species, so that the size of the nuclear genome can be applied to study the phylogenetic and systematic relationship of many taxonomic groups [49]. According to the DNA amounts of samples, there was a significant difference in DNA content of Turkey and Iran’s diploid and tetraploid *Aegilops* species (*p* ˂ 0.01); however, in hexaploid species, the significant difference was not observed. This was not surprising since significant DNA content differences were observed among base genomes (diploid species). There was high discrepancy between the results of this study and the DNA C-values at the Royal Botanic Garden database for *Ae. cylindrica* and *Ae. triaristata*. This is probably due to applied methods since most values at the Royal Botanic Garden database are based on Feulgen Micro spectrophotometry, which is an outdated method. However, our results were very similar to other findings [26,31,42], which were based on flow cytometry. Despite the large variation at the interspecific level, great stability could be observed at the intraspecific level and all species with the same genomic constitution had similar DNA content. The results thus support conclusions of others researchers [42,50,51,52,53]. Our results were also consistent with Lee et al. [54], which used flow cytometry and the chromosome imaging technique in hexaploid wheat, hexaploid triticale, tetraploid wheat and AA, BB, DD genome donors. 

## 5. Conclusions

The variation of nuclear DNA content in diploid, tetraploid as well as hexaploid *Aegilops* natural populations was studied. Based on results, we found that the mean nuclear DNA contents were significant in diploid *Aegilops* and tetraploid populations, but it was not in hexaploid *Aegilops* in both Turkey and Iran’s geographic conditions. In addition, the nuclear DNA content variations were high in all the studied ploidy levels: 0.86 pg/2C and 1.36 pg/2C at diploid level, 3.56 pg/2C and 3.14 pg/2C at tetraploid level, 0.22 pg/2C and 1.39 pg/2C at hexaploid level in Turkey and Iran’s populations, respectively. The maximum (3.56 pg/2C) and minimum (0.22 pg/2C) nuclear DNA content were observed in tetraploid and hexaploid populations in Turkey. Thus, it was assumed that these large variations in nuclear DNA content might be correlated with geographical or climate parameters. It could also be concluded that the flow cytometry is a valuable technique for basic and applied studies whose new applications continue to emerge in several areas. In addition, in the genus *Aegilops*, identification and validation for its many varied species based solely on morphological features may not be sufficient and can be difficult. Therefore, determination of relative nuclear DNA values by flow cytometry could serve as a supplementary simple and routine method for identification and maintenance of accessions. Our results also proved that this technique could provide fast and highly reliable determination of *Aegilops* ploidy surface.

## Figures and Tables

**Table 1 genes-13-01980-t001:** Ecotypes of *Aegilops* species collected from Turkey and Iran.

Turkey		Iran	
Species	Ecotypes	Species	Ecotypes
*Ae. biuncialis*	Adiyaman	*Ae. biuncialis*	Urmia
*Ae. biuncialis*	Ankara	*Ae. biuncialis*	Rasht
*Ae. biuncialis*	Kahraman Marash	*Ae. biuncialis*	Saari
*Ae. biuncialis*	Malatya	*Ae. biuncialis*	Kerman
*Ae. biuncialis*	Gerede	*Ae. biuncialis*	Shiraz
*Ae. columnaris*	Adiyaman	*Ae. columnaris*	Neyriz
*Ae. columnaris*	Ankara	*Ae. columnaris*	Khoramabad
*Ae. columnaris*	Nevsehir	*Ae. columnaris*	Saghez
*Ae. columnaris*	Van	*Ae. columnaris*	Zahedan
*Ae. columnaris*	Denizli	*Ae. columnaris*	Zanjan
*Ae. crassa*	Adiyaman	*Ae. crassa*	Gazvin
*Ae. crassa*	Ankara	*Ae. crassa*	Aligudarz
*Ae. crassa*	Kirikkale	*Ae. crassa*	Shiraz
*Ae. crassa*	Tufanbeyli	*Ae. crassa*	Ilam
*Ae. crassa*	Konya	*Ae. crassa*	Marvdasht
*Ae. cylindrica*	Ankara	*Ae. cylindrica*	Semnan
*Ae. cylindrica*	Konya	*Ae. cylindrica*	Tabriz
*Ae. cylindrica*	Kirikkale	*Ae. cylindrica*	Mashhad
*Ae. cylindrica*	Haymana	*Ae. cylindrica*	Shiraz
*Ae. cylindrica*	Igdir	*Ae. cylindrica*	Ghorghan
*Ae. speltoides* var. *aucheri*	Ankara	*Ae. speltoides* var. *aucheri*	Saari
*Ae. speltoides* var. *aucheri*	Konya	*Ae. speltoides* var. *aucheri*	Ilam
*Ae. speltoides* var. *aucheri*	Sanliurfa	*Ae. speltoides* var. *aucheri*	Zahedan
*Ae. speltoides* var. *aucheri*	Corum	*Ae. speltoides* var. *aucheri*	Arak
*Ae. speltoides* var. *aucheri*	Mus	*Ae. speltoides* var. *aucheri*	Sanandaj
*Ae. speltoides* var. *ligustica*	Konya	*Ae. speltoides* var. *ligustica*	Kermanshah
*Ae. speltoides* var. *ligustica*	Erzurum	*Ae. speltoides* var. *ligustica*	Ghuchan
*Ae. speltoides* var. *ligustica*	Igdir	*Ae. speltoides* var. *ligustica*	Zanjan
*Ae. speltoides* var. *ligustica*	Corum	*Ae. speltoides* var. *ligustica*	Shiraz
*Ae. speltoides* var. *ligustica*	Karaman	*Ae. speltoides* var. *ligustica*	Saari
*Ae. tauschii*	Kirsehir	*Ae. tauschii*	Shiraz
*Ae. tauschii*	Konya	*Ae. tauschii*	Yasuj
*Ae. tauschii*	Sanliurfa	*Ae. tauschii*	Khorasan
*Ae. tauschii*	Van	*Ae. tauschii*	Urmia
*Ae. tauschii*	Kahramanmaras	*Ae. tauschii*	Zahedan
*Ae. triaristata*	Adana	*Ae. triuncialis*	Hamedan
*Ae. triaristata*	Osmaniye	*Ae. triuncialis*	Ahvaz
*Ae. triaristata*	Tufanbeyli	*Ae. triuncialis*	Lahijan
*Ae. triaristata*	Aydin	*Ae. triuncialis*	Kermanshah
*Ae. triaristata*	Manisa	*Ae. triuncialis*	Shiraz
*Ae. triuncialis*	Adiyaman	*Ae. umbellulata*	Tabriz
*Ae. triuncialis*	Batman	*Ae. umbellulata*	Esfahan
*Ae. triuncialis*	Siirt	*Ae. umbellulata*	Ghonbad
*Ae. triuncialis*	Van	*Ae. umbellulata*	Ardebil
*Ae. triuncialis*	Bitlis	*Ae. umbellulata*	Kerman
*Ae. umbellulata*	Mardin		
*Ae. umbellulata*	Nuseybin		
*Ae. umbellulata*	Denizli		
*Ae. umbellulata*	Selcuk		
*Ae. umbellulata*	Mus		

**Table 2 genes-13-01980-t002:** Somatic chromosomes number, mean total chromosome length, mean arm ratio and karyotypic formula of *Aegilops* species in Iran’s ecotypes.

Species	2n	MTCL ± Se (µm)	AR + Se	KF
*Ae. biuncialis*	28	108.32 ± 0.63	2.54 ± 0.92	10sm + 4st + 3sat
*Ae. columnaris*	28	103 ± 0.21	2.12 ± 0.29	10sm + 3m + 1st + 2sat
*Ae. crassa* (4x)	28	156 ± 0.21	1.46 ± 0.03	11m + 3sm + 2sat
*Ae. crassa* (6x)	42	217.39 ± 0.14	1.36 ± 0.03	20m + 1sm + 3sat
*Ae. cylindrica* (4x)	28	105.97 ± 0.19	2.37 ± 0.53	8sm + 5st + 1m + 1sat
*Ae. cylindrica* (6x)	42	237 ± 0.12	2.29 ± 0.19	12sm + 8st + 2m + 2sat
*Ae. speltoides* Var. Aucheri	14	56.22 ± 0.37	1.37 ± 0.04	6m + 1sm + 1sat
*Ae. speltoides* Var. Ligustica	14	82.12 ± 0.17	1.52 ± 0.22	5sm + 2m + 5sat
*Ae. speltoides* Var. Taush.	14	63.18 ± 0.33	1.38 ± 0.18	7m
*Ae. triuncialis*	28	108.52 ± 0.16	2.45 ± 0.91	14st
*Ae. umbellulata*	28	25.95 ± 0.09	2.84 ± 0.17	5st + 1sm + 1t + 1sat

MTCL: Mean Total Chromosome Length, Se: Standard error, µm: Micrometer, AR: Arm Ratio, KF: karyotype formula, Sat: Satellite, m: Metacentric, sm: Submetacentric, st: Sub telocentric, t: Telocentric.

**Table 3 genes-13-01980-t003:** Somatic chromosomes number, mean total chromosome length, mean arm ratio and karyotypic formula of *Aegilops* species in Turkey’s ecotypes.

Species	2n	MTCL	AR	KF
*Ae. biuncialis*	28	98.69 ± 0.61	2.15 ± 0.69	13sm + 1m + 3sat
*Ae. columnaris*	28	103.32 ± 0.79	2.12 ± 1.17	10sm + 3m + 1st + 2sat
*Ae. crassa* (4x)	28	153.35 ± 0.49	1.61 ± 0.05	11m + 3sm + 2sat
*Ae. crassa* (6x)	42	206.11 ± 0.03	1.13 ± 0.04	10m + 1sm + 3sat
*Ae. cylindrica* (4x)	28	81.57 ± 0.10	2.39 ± 0.21	8sm + 5st + 1m + 1sat
*Ae. cylindrica* (6x)	42	245 ± 0.08	2.30 ± 0.12	12sm + 8st + 2m + 2sat
*Ae. speltoides* Var. Aucheri	14	52 ± 0.12	1.31 ± 0.09	7m + 1sat
*Ae. speltoides* Var. Ligustica	14	69.35 ± 0.15	1.67 ± 0.19	5sm + 2m + 5sat
*Ae. speltoides* Taush.	14	56.10 ± 0.02	1.47 ± 0.24	7m
*Ae. triuncialis*	28	108.32 ± 0.01	3.81 ± 1.63	12st + 2sm
*Ae. triaristata*	28	81.99 ± 0.10	2.17 ± 0.05	9sm + 5m + 1B chr
*Ae. umbellulata*	14	29.21 ± 0.62	2.94 ± 0.08	3sm + 2st + 2t + 1sat

MTCL: Mean Total Chromosome Length, Se: Standard error, µm: Micrometer, AR: Arm Ratio, KF: karyotype formula, Sat: Satellite, m: Metacentric, sm: Submetacentric, st: Subtelocentric, t: Telocentric.

**Table 5 genes-13-01980-t005:** The nuclear DNA content among *Aegilops* ecotypes.

	Mean of Square					
	Turkey			Iran		
S.O.V	D	T	H	D	T	H
Plant Species	0.548 **	2.861 **	0.096 ^ns^	1.222 **	5.328 **	1.444 ^ns^
Error	0.119	0.227	0.104	0.184	0.593	0.995

**: Significant at *p* < 0.01; ^ns^: Not significant, D: Diploid; T: Tetraploid; H: Hexaploid.

**Table 6 genes-13-01980-t006:** Somatic chromosomes number, mean total chromosome length, mean arm ratio, karyotypic formula and values of the nuclear DNA content of *Aegilops* species in Iran’s ecotypes.

Species	2n	MTCL ± Se (µm)	AR + Se	KF	DNA Content ± Se	Genome
*Ae. biuncialis*	28	108.32 ± 0.63	2.54 ± 0.92	10sm + 4st + 3sat	20.23 ± 0.35	UUMM
*Ae. olumnaris*	28	103 ± 0.21	2.12 ± 0.29	10sm + 3m + 1st + 2sat	21.75 ± 0.36	UUMM
*Ae. crassa* (4x)	28	156 ± 0.21	1.46 ± 0.03	11m + 3sm + 2sat	20.44 ± 0.29	DDMM
*Ae. crassa* (6x)	42	217.39 ± 0.14	1.36 ± 0.03	20m + 1sm + 3sat	33.97 ± 0.06	DDMM
*Ae. cylindrica* (4x)	28	105.97 ± 0.19	2.37 ± 0.53	8sm + 5st + 1m + 1sat	18.61 ± 0.07	CCDD
*Ae. cylindrica* (6x)	42	237 ± 0.12	2.29 ± 0.19	12sm + 8st + 2m + 2sat	32.58 ± 0.25	CCDD
*Ae. speltoides* var. Aucheri	14	56.22 ± 0.37	1.37 ± 0.04	6m + 1sm + 1sat	10.22 ± 0.02	SS
*Ae. speltoides* var. Ligustica	14	82.12 ± 0.17	1.52 ± 0.22	5sm + 2m + 5sat	11.56 ± 0.01	SS
*Ae. speltoides* var. Taush.	14	63.18 ± 0.33	1.38 ± 0.18	7m	10.20 ± 0.36	SS
*Ae. triuncialis*	28	108.52 ± 0.16	2.45 ± 0.91	14st	19.27 ± 0.04	UUCC
*Ae. umbellulata*	28	25.95 ± 0.09	2.84 ± 0.17	5st + 1sm + 1t + 1sat	10.58 ± 0.56	UU

MTCL: Mean Total Chromosome Length, Se: Standard error, µm: Micrometer, AR: Arm Ratio, KF: karyotype formula, Sat: Satellite, m: Metacentric, sm: Submetacentric, st: Sub telocentric, T: telocentric.

**Table 7 genes-13-01980-t007:** Somatic chromosomes number, mean total chromosome length, mean arm ratio, karyotypic formula and values of the nuclear DNA content of *Aegilops* species in Turkey’s ecotypes.

Species	2n	MTCL ± Se (µm)	AR ± Se	KF	DNA Content ± Se	Genome
*Ae. biuncialis*	28	98.69 ± 0.61	2.15 ± 0.69	13sm + 1m + 3sat	19.89 ± 0.29	UUMM
*Ae. columnaris*	28	103.32 ± 0.79	2.12 ± 1.17	10sm + 3m + 1st + 2sat	20.17 ± 0.16	UUMM
*Ae. crassa* (4x)	28	153.35 ± 0.49	1.61 ± 0.05	11m + 3sm + 2sat	20.08 ± 0.01	DDMM
*Ae. crassa* (6x)	42	206.11 ± 0.03	1.13 ± 0.04	10m + 1sm + 3sat	31.59 ± 0.03	DDMM
*Ae. cylindrica* (4x)	28	81.57 ± 0.10	2.39 ± 0.21	8sm + 5st + 1m + 1sat	18.09 ± 0.05	CCDD
*Ae. cylindrica* (6x)	42	245 ± 0.08	2.30 ± 0.12	12sm + 8st + 2m + 2sat	31.81 ± 0.08	CCDD
*Ae. speltoides* var. Aucheri	14	52 ± 0.12	1.31 ± 0.09	7m + 1sat	10.15 ± 0.04	SS
*Ae. speltoides* var. Ligustica	14	69.35 ± 0.15	1.67 ± 0.19	5sm + 2m + 5sat	10.95 ± 0.12	SS
*Ae. speltoides* Taush.	14	56.10 ± 0.02	1.47 ± 0.24	7m	10.16 ± 0.27	SS
*Ae. triuncialis*	28	108.32 ± 0.01	3.81 ± 1.63	12st + 2sm	19.05 ± 0.03	UUCC
*Ae. triaristata*	28	81.99 ± 0.10	2.17 ± 0.05	9sm + 5m + 1B chr	21.65 ± 0.05	UUMM
*Ae. umbellulata*	14	29.21 ± 0.62	2.94 ± 0.08	3sm + 2st + 2t + 1sat	10.09 ± 0.04	UU

MTCL: Mean Total Chromosome Length, Se: Standard error, µm: Micrometer, AR: Arm Ratio, KF: karyotype formula, Sat: Satellite, m: Metacentric, sm: Submetacentric, st: Subtelocentric, t: Telocentric.

## Data Availability

The original contributions presented in the study are included in the article, further inquiries can be directed to the corresponding authors.

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
