# Peer review of "Estimation of Nuclear DNA Content in Some *Aegilops* Species: Best Analyzed Using Flow Cytometry"

_genes, 2022, doi:10.3390/genes13111980_

Round 1
Reviewer 1 Report
This this manuscript, authors measured genome size (nuclear DNA content) of collected accessions for each of several Aegilops species of diploid, tetraploid and hexaploid levels from Turkey and Eran. In parallel, thy did cytogenetic analysis for these samples. Major findings include larger differences across species of different ploidies, larger inter- than intra-specific variation in genome size, and notable intraspecific variation among accession collected from the two nations. Compared with existing publications in this topic about wheat and Aegilops, novelty of these results is limited. More troublesome is the numerous mistakes and inaccuracies throughout the manuscript. I will list only a few examples. To conclude, unless more data can be added and the manuscript thoroughly revised, I do not recommend its publication.
1. The tile is already misleading: “Diversity of Nuclear DNA Contents of Wild Wheat Variety…”. First, authors only studied species of Aegilops and none of Triticum. Second, “variety” usually refers to a cultivar not an accession from the wild.
2. Line 16: “cytogenetic diversity” is misleading, because authors mainly measured nuclear DNA contents; the cytogenetic data shown are not informative to reflect cytogenetic diversity. For example, structural chromosomal variation, like translocation and inversion, which his th most imorant information about cytogenetic diversity not analyzed.
3. Lines 36-37: “…the size of genome in wild self-fertilizing wheat species is generally stable.”. first, wheat was not studied. Second, Aegilops speltoides is well-known as a outcrosser.
Author Response
Reviewer 1
This this manuscript, authors measured genome size (nuclear DNA content) of collected accessions for each of several Aegilops species of diploid, tetraploid and hexaploid levels from Turkey and Eran. In parallel, thy did cytogenetic analysis for these samples. Major findings include larger differences across species of different ploidies, larger inter- than intra-specific variation in genome size, and notable intraspecific variation among accession collected from the two nations. Compared with existing publications in this topic about wheat and Aegilops, novelty of these results is limited. More troublesome is the numerous mistakes and inaccuracies throughout the manuscript. I will list only a few examples. To conclude, unless more data can be added and the manuscript thoroughly revised, I do not recommend its publication.
Authors’ response: Thanks for your very deep consideration and comments. You are right. I had so many data regarding the cytological studies on the studied Aegilops species mentioned in this manuscript. I wanted to include them in another paper but now after your comments regarding inadequate data, I will add some of them in the revised manuscript. Hope that it will be enough for this paper. But if you think that these data may be inadequate, I can add the all the chromosomes analysis tables in this manuscript.
- The tile is already misleading: “Diversity of Nuclear DNA Contents of Wild Wheat Variety…”. First, authors only studied species of Aegilops and none of Triticum. Second, “variety” usually refers to a cultivar not an accession from the wild.
Authors’ response: The title of the manuscript is now changed per the suggestion.
- Line 16: “cytogenetic diversity” is misleading, because authors mainly measured nuclear DNA contents; the cytogenetic data shown are not informative to reflect cytogenetic diversity. For example, structural chromosomal variation, like translocation and inversion, which his th most imorant information about cytogenetic diversity not analyzed.
Authors’ response: The cytogenetic diversity was changed to the cytogenetic analysis in the suggested line. Also, some data results were added to the result section including karyotype analysis of all studied species.
- Lines 36-37: “…the size of the genome in wild self-fertilizing wheat species is generally stable.”. first, wheat was not studied. Second, Aegilops speltoides is well-known as a outcrosser.
Authors’ response: The sentence “the size of the genome in wild self-fertilizing wheat species is generally stable.” in line 36-37 were deleted in the revised manuscript.
4) The respected reviewer mentioned that the introduction, references, methods, as well as results should be improved. Also, there is needed to be English edited extensively.
Authors’ response: The mentioned sections were improved, and new references were also added to the revised manuscript. Also, the manuscript was extensively English-edited.
Reviewer 2 Report
This research has a great interest for the genetic resources communities and the understanding of the genome variation of wild species. This is very important for both the conservation (including collecting) but also the use of these species in interspecific crosses and their crossability with other species. High genotyping techniques could also be done to further study these species and give more insights to users of these materials. Congratulations for this work.
Line45: please specify which kind of adaptation to water. Water scarcity? Different water regimes?
Line54: no need to add “for breeders”
Line54: the sentence “It was reported …” does not fit into the paragraph or a liaison sentence is missing, please improve.
Lines63: replace belonged to belonging
Line67: protein quality or protein content?
Lines 130-131: Please rephrase “Morphological characterization” and DNA extraction
Table1: Passport data such as coordinates would be of great importance for readers and users
Line 167: reference for the formula
Line 227: rephrase the sentence as it does not make a lot of sense and the sentence may be placed in the results paragraph
In M&M, it is not clear how sampling for collecting was done. Is it based on gap analysis, eco-geographic surveys…?
Author Response
Reviewer 2
- This research has a great interest for the genetic resources communities and the understanding of the genome variation of wild species. This is very important for both the conservation (including collecting) but also the use of these species in interspecific crosses and their cross ability with other species. High genotyping techniques could also be done to further study these species and give more insights to users of these materials. Congratulations for this work.
Authors’ response: We are very much Thankful to the reviewer for appreciating our work
- Line45: please specify which kind of adaptation to water. Water scarcity? Different water regimes?
Authors’ response: The necessary information was added to the text (Line 45) with new references in the revised manuscript.
- Line54: no need to add “for breeders”
Authors’ response: It was replaced
- Line54: the sentence “It was reported …” does not fit into the paragraph or a liaison sentence is missing, please improve.
Authors’ response: It was deleted in the revised manuscript
- Lines 63: replace belonged to belonging
Authors’ response:
- Line67: protein quality or protein content?
Authors’ response: corrected as protein content
- Lines 130-131: Please rephrase “Morphological characterization” and DNA extraction
Authors’ response: corrected. Line 143
- Line 167: reference for the formula
Authors’ response: Mentioned. Line 182
- Line 227: rephrase the sentence as it does not make a lot of sense and the sentence may be placed in the results paragraph
Authors’ response: Rephrased
- In M&M, it is not clear how sampling for collecting was done. Is it based on gap analysis, eco-geographic surveys…?
Authors’ response: It was based on eco-geographic surveys. Mentioned at Line 134

Round 2
Reviewer 1 Report
After a careful reassessment, I fell the manuscript has been substantially improved. In particular, authors have adequately amended my major concerns raised to the earlier version. The added cytogenetic data are adequate and absolutely necessary. Presentation and language are also improved. These said, I still suggest do another round of careful-proofreading to ensure absence of typos and grammar mistakes.